# Macrophages Orchestrate Airway Inflammation, Remodeling, and Resolution in Asthma

**DOI:** 10.3390/ijms241310451

**Published:** 2023-06-21

**Authors:** Rodney D. Britt, Anushka Ruwanpathirana, Maria L. Ford, Brandon W. Lewis

**Affiliations:** 1Center for Perinatal Research, Abigail Wexner Research Institute at Nationwide Children’s Hospital, Columbus, OH 43215, USA; rodney.britt@nationwidechildrens.org (R.D.B.J.); anushka.ruwanpathirana@nationwidechildrens.org (A.R.); maria.ford@nationwidechildrens.org (M.L.F.); 2Department of Pediatrics, The Ohio State University, Columbus, OH 43210, USA; 3Biomedical Sciences Graduate Program, College of Medicine, The Ohio State University, Columbus, OH 43205, USA

**Keywords:** macrophages, asthma, inflammation, remodeling

## Abstract

Asthma is a heterogenous chronic inflammatory lung disease with endotypes that manifest different immune system profiles, severity, and responses to current therapies. Regardless of endotype, asthma features increased immune cell infiltration, inflammatory cytokine release, and airway remodeling. Lung macrophages are also heterogenous in that there are separate subsets and, depending on the environment, different effector functions. Lung macrophages are important in recruitment of immune cells such as eosinophils, neutrophils, and monocytes that enhance allergic inflammation and initiate T helper cell responses. Persistent lung remodeling including mucus hypersecretion, increased airway smooth muscle mass, and airway fibrosis contributes to progressive lung function decline that is insensitive to current asthma treatments. Macrophages secrete inflammatory mediators that induce airway inflammation and remodeling. Additionally, lung macrophages are instrumental in protecting against pathogens and play a critical role in resolution of inflammation and return to homeostasis. This review summarizes current literature detailing the roles and existing knowledge gaps for macrophages as key inflammatory orchestrators in asthma pathogenesis. We also raise the idea that modulating inflammatory responses in lung macrophages is important for alleviating asthma.

## 1. Introduction

Macrophages are important immune cells and present in virtually every tissue in the body [1]. Lung macrophages contribute to lung homeostasis by patrolling airways and removing dead cells, inhaled particles, and foreign invaders such as bacteria [2,3]. Macrophages regulate immune responses through cytokine production and their pathogen killing capabilities are essential for clearing foreign invaders such as viruses and bacteria away from airways [4]. Additionally, macrophages are important players in the resolution of pulmonary inflammation and wound healing processes [5,6,7]. It may be possible to therapeutically target macrophages to reduce their deleterious effects, for example inflammatory mediator production, while simultaneously maintaining or even enhancing the beneficial effects of macrophages in host defense, immune suppression, and resolution. Given their importance in driving immune responses, structural integrity, and host defense, it is critical to understand the roles macrophages play in chronic inflammatory lung diseases.

Asthma is a chronic inflammatory lung disease characterized by increased immune cell infiltration, airway thickening or remodeling, and airway hyperresponsiveness leading to restricted airflow and difficulty breathing [8]. Asthma phenotypes/endotypes vary from a predominant type 2 response with increased T helper (Th) 2 cell populations and interleukin (IL)-4, -5, -13 levels to more severe endotypes that have additional presence of type 1 and/or type 17 inflammation, which are associated with corticosteroid insensitivity [9]. People with severe asthma experience more frequent exacerbations and account for an estimated 40–70% of the health care costs associated with asthma [10]. In addition to type 1/17 inflammation, severe asthma is often associated with mixed granulocytic eosinophil and neutrophil lung infiltration [11]. The presence of these mixed granulocytic immune cell populations is associated with decreased responsiveness to current corticosteroid therapies [11,12,13]. Notably, macrophages contribute to pathways that promote type 1, 2, and 17 inflammation, indicating a possible role in contributing to asthma phenotypes/endotypes and severity [14,15,16].

Macrophages are key orchestrators of the immune response by recruiting eosinophils, neutrophils, and monocytes, as well as activate effector Th cells that further enhance inflammation [17,18,19,20,21]. Persistent airway thickening and remodeling consists of increased airway epithelial thickening, mucus hypersecretion, airway smooth muscle mass, and collagen deposition leading to stiffening of the airway and restriction of airflow [3,22,23,24,25]. Macrophages secrete factors that promote airway remodeling, including IL-4 and IL-13, and profibrotic growth factors, transforming growth factor-β (TGF-β) and platelet derived growth factor (PDGF) [26,27]. Given the importance of macrophages in mediating immune responses, controlling infection, and initiating remodeling/resolution, targeting these important cells may provide beneficial therapeutics for asthma (Figure 1). This review will discuss current understanding of macrophages and their diverse roles in asthma. We also discuss prospective avenues and opportunities for further exploration into the role of lung macrophages in asthma.

## 2. Lung Macrophage Subsets

Lung macrophage populations are heterogenous and consist of multiple subsets with varied localization within the lung, functions, and transcriptional profiles. They are identified by surface marker expression, location, and origin: alveolar, interstitial, and recruited (monocyte-derived) [28,29]. Each subpopulation shares similar surface markers, such as CD45+, CD64+, and F4/80+, but are distinguished by additional surface markers [30,31]. In this section, we will discuss markers used to identify subsets and the known functions associated with each subset.

### 2.1. Alveolar Macrophages

Alveolar macrophages are located in the lung airspace within alveoli and are critical in maintaining homeostasis in the lung, which is constantly bombarded with inhaled pathogens, particles, and noxious gases [32]. The alveolar macrophage population arises from fetal monocytes that infiltrate the lung during fetal development and undergo differentiation into alveolar macrophages through paracrine signaling from alveolar epithelial cells [33]. This resident population is self-renewing and maintained by proliferation, with little contribution from circulating monocytes [33,34]. Alveolar macrophages are identified by CD64+, CD11b−, major histocompatibility complex (MHC) II+, CD11c+, MerTK+, and Siglec F+ expression [30,31]. They act as a first line of defense to environmental insults and play a central role in maintaining lung homeostasis by regulating removal of bacterial pathogens and particles via phagocytosis, surfactant maintenance, and tissue repair [5,35,36,37].

### 2.2. Interstitial Macrophages

Interstitial lung macrophages are another lung resident macrophage population that reside within interstitial areas around peri-bronchiolar, peri-vascular spaces, and alveolar walls [2,38,39]. Interstitial lung macrophage populations also have a fetal origin but can be replenished by circulating monocytes that infiltrate the lung and differentiate into macrophages [2]. Interstitial macrophages are identified by CD64+, CD11b+, CD11c+, MHCII+, MerTK+, and Siglec F- expression [31]. This macrophage subset has key roles in inflammation, tissue repair and fibrosis, and antigen presentation [40]. The function of interstitial macrophages is somewhat distinct from alveolar macrophages as they lack the ability to support surfactant production and maintenance [41]. Their close proximity to the airway and vasculature enables them to affect the surrounding environment by secreting cytokines, chemokines, and growth factors [42,43,44]. Their MHCII expression suggests a key role in antigen presentation and adaptive immune cell activation [44]. Their immunoregulatory functions include IL-10 secretion, a cytokine known to suppress inflammation by inhibiting IFN-γ, TNF-α, and IL-5 production [37,43,45].

### 2.3. Recruited Macrophages

Upon acute lung inflammation, monocytes are recruited from the bone marrow or circulation and infiltrate into the lung. For recruited monocytes/monocyte-derived macrophages, surface marker expression changes once monocytes enter the lung and become macrophages [46]. Recruited monocytes are identified as MHCII-, CD11b+, Ly6C+, and by low CD64 expression [31]. Once monocytes enter the lung and differentiate into monocyte-derived macrophages, these cells are identified as CD64+, CD11c+, F4/80+, MerTK+, and low Siglec F expression [31]. The C-C motif chemokine ligand (CCL2) and its receptor CCR2 is a well characterized chemoattractant axis for monocyte recruitment [47]. Monocytes produce high amounts of pro-inflammatory cytokines and chemokines that contribute to immune cell recruitment and activation. They also differentiate into both alveolar and interstitial macrophages, helping replenish resident lung macrophage populations after an inflammatory event [37].

Taken together, there is likely to be a division of labor between lung macrophage subsets [37]. Each population has important roles in driving innate and adaptive immune mechanisms in lung inflammation. For asthma, their differential roles are just beginning to be revealed in mouse models of allergic airway inflammation and ex vivo studies from human lung macrophages.

### 2.4. Macrophage Activation

Macrophage activation status is identified using various markers to determine classical or alternative activation. Classical macrophage activation is induced by type 1 cytokines such as IFN-γ, and bacterial products such as lipopolysaccharide [48]. These macrophages produce type 1 cytokines: IL-6, IL-1β, and TNF-α [49]. Classically-activated macrophages are also identified by enhanced inducible nitric oxide synthase (iNOS) expression [49]. Alternative macrophage activation is induced by type 2 cytokines, IL-4 and IL-13 [48,50]. In response, alternatively-activated macrophages produce IL-13, IL-5, TGFβ, and IL-10 [50]. In mice, alternatively-activated macrophages can be identified by Fizz1 or arginase (Arg-1) [51]. In humans, CD206 and MARCO are being used as markers for alternative macrophage activation [52]. In summary, macrophages readily respond to the inflammatory environment in order to secrete mediators that drive inflammation. In depth analyses does not need to focus solely on location of the macrophages but also macrophage activation status, in order to truly understand their role in airway diseases such as asthma.

## 3. Role of Macrophages in Asthma Pathogenesis

Asthma is a heterogenous disease that involves complex mechanisms between immune cells and airway structural cells. Lung macrophages and monocytes contribute to allergic airway inflammation by promoting inflammatory cell recruitment/activation and secreting factors that induce structural cell thickening and remodeling. Macrophages and monocytes also play important roles in inflammatory resolution post-exacerbation by suppressing immune responses and contributing to tissue repair processes [53]. In this section, we discuss the roles of lung macrophages and monocytes in asthma pathogenesis.

### 3.1. Eosinophils and Neutrophils

Eosinophils are granulocytes that have long been associated with asthma and type 2 associated responses and serve as a key biomarker for asthma phenotypes/endotypes [54]. They are recruited and activated via mechanisms mediated by CCL11, CCL24, and IL-5 among other type 2 cytokines and chemokines [55,56]. Eosinophils release granules containing IL-4 and IL-10 that further enhance type 2 immune responses commonly associated with asthma [57]. Biologic therapies for asthma include anti-IL-5 (mepolizumab) and anti-IL-5R (benralizumab), which neutralize IL-5 signaling [58]. These targeted therapies reduce circulating eosinophils in asthma patients and reduce exacerbation frequency, highlighting the importance of eosinophils and type 2 inflammation in asthma pathogenesis [59,60].

Macrophages play an important role in recruiting eosinophils from the bone marrow to the site of inflammation and activation. Eosinophil recruitment and migration to the lung is driven by CCR3, which binds chemokines CCL11 and CCL24 [61]. In an ovalbumin mouse model, isolated alveolar macrophages were shown to secrete CCL24, while interstitial lung macrophages secreted CCL11 at greater levels [62]. It was proposed that CCL11 attracts circulating eosinophils, while CCL24 guides eosinophils to infiltrate airspaces, suggesting distinct contributions from resident lung macrophage populations [62,63]. Recruited monocytes have also been shown to contribute to eosinophilic infiltration. Intravenous clodronate administration to deplete circulating monocytes decreased eosinophil numbers in the BAL of HDM challenged mice [64]. However, less is known about how circulating monocytes contribute to eosinophil recruitment to the lung. In vitro studies in THP-1 cells, a human monocyte cell line, stimulated with PMA, IL-4, and IL-13 showed increased CCL24 production [65]. In summary, each macrophage subset has been linked to the recruitment of eosinophils, thereby displaying potential to enhance type 2 inflammatory responses.

While not part of type 2 allergic responses, the presence of neutrophils is strongly linked to severe asthma and reduced response to corticosteroid treatment [66]. These cells are recruited from the bone marrow to sites of inflammation through IL-6 and CXCL1 [67,68]. IL-6 stimulates neutrophil release from the bone marrow [67] and CXCL1 serves as a key neutrophil chemoattractant via CXCR2 [69]. Bronchoalveolar lavage (BAL) macrophages from severe asthma patients exhibit increased gene expression in *IL6* and *CXCL1*, among other pro-inflammatory mediators, indicating a role of neutrophil recruitment and activation by macrophages in humans [70]. Alveolar macrophages recruit and stimulate neutrophils by secreting granulocyte-colony stimulating factor (G-CSF), IL-6, and CXCL1 in mouse models of allergic airway disease [62,71] and other lung disease models, such as cystic fibrosis [72]. G-CSF is important for neutrophil maturation in the bone marrow and its expression is increased in neutrophilic asthma [73]. Interstitial macrophages have been shown to suppress neutrophil infiltration by secreting IL-10. HDM-challenged *Il10^−/−^* mice exhibited increased neutrophil infiltration compared to HDM-challenged wild type (WT) mice. Adoptive transfer of WT interstitial macrophages into HDM-challenged *Il10^−/−^* mice reduced neutrophil infiltration, indicating a protective role for interstitial lung macrophages [43]. Recruited monocytes have been identified as producing IL-6 in HDM-challenged mice [74]. These cells are also identified as major recruiters of neutrophils in cystic fibrosis through CXCL1 production [24]. Overall, macrophage subsets have different roles in neutrophilic inflammation by either recruitment or suppression. It will be important to further understand how macrophages affect neutrophil infiltration and consequently asthma severity.

### 3.2. Monocytes

Monocytes are found in higher numbers in sputum from asthma patients compared to healthy controls and contribute to increased inflammation in mice with allergic airway inflammation [64,75]. Monocytes arise from the bone marrow and are recruited to the lung during inflammation through the CCL2/CCR2 recruitment axis. CCL2 expression has been found to be increased in asthma patient BAL compared to healthy controls, suggesting enhanced monocyte recruitment in asthma [76,77]. In allergen provocation in humans, CCL2 is rapidly induced and followed by subsequent monocyte-derived macrophage infiltration to the airway [77]. Although CCL2 is produced by other cell types, such as airway epithelial cells, recent studies have found lung macrophages to be a major source of CCL2, implicating them as orchestrators of monocyte recruitment [78,79]. In a mouse model of lung injury, alveolar macrophages were shown to be major producers of CCL2, as depletion of alveolar macrophages via clodronate reduced CCL2 levels in the lung [80].

The type 2 cytokine, IL-13, is important for increased CCL2 expression in allergen-challenged mice [81]. Adoptive transfer of IL-4Rα, the receptor for IL-13, expressing macrophages into ovalbumin sensitized and challenged mice enhanced allergic airway inflammation that included substantial increases in CCL2 and CCL11 BAL expression, suggesting that macrophages promote CCL2 production [82]. A recent study found high CCL2 expression in alternatively activated macrophages and implicated CCL2-CCR2 signaling in macrophage activation through a mechanism via miR-511-3p targeting, binding, and reducing CCL2 expression [83]. Deletion of *Mrc1*, where miR-511-3p is encoded, in mice heightened airway inflammation and hyperresponsiveness while enhancing macrophage activation [83]. Collectively, these studies suggest that CCL2 production by lung macrophages contributes to monocyte recruitment and macrophage activation in asthma.

### 3.3. Innate Lymphoid Cells

Innate lymphoid cells (ILCs) are an important lymphocyte population that are key sources of type 1, 2, and 17 effector cytokines in asthma. They are distinguished from T lymphocytes by their lack of requirements for receptor recombination [84]. Type 2 ILCs (ILC2) are stimulated by type 2-associated cytokines, IL-33, IL-25, and thymic stromal lymphopoietin (TSLP) to differentiate and secrete IL-5 and IL-13 [85]. In addition to airway epithelial cells, macrophages express and secrete IL-33 to enhance type 2 inflammation in allergic airway inflammation [86,87,88]. Macrophages also secrete IL-4 and IL-13 to further enhance ILC2 responses [89,90]. Myeloid IL-4Rα deficiency in mice shows reduced type 2 inflammation and ILC2 populations [91], implicating macrophage IL-4Rα signaling as an important contributor to ILC2 activation. Pro-inflammatory signaling in macrophages may also antagonize ILC2 expansion and type 2 inflammation. Stimulator of IFN genes (STING) activation by 2′3′-cGAMP in alveolar macrophages induces IRF3-type I IFN signaling which inhibits IL-33-induced ILC2 activation [92]. In interstitial lung macrophages, toll-like receptor (TLR) 7 activation was recently shown to induce IL-27 production and suppress ILC2 populations and type 2 inflammation in IL-33-induced eosinophilic airway inflammation in mice [93]. More in-depth investigation into macrophage enhancement of ILC2 cytokine secretion may provide possible therapeutic targets to mitigate enhanced type 2 responses.

In type 1 inflammation, macrophages are major producers of IL-12 and IL-18 in asthma [94,95]. Similar to Th1 cells, ILC1s respond to IL-12 and IL-18 to secrete IFN-γ and TNF-α [96,97,98]. IFN-γ and TNF-α have been found to increase corticosteroid insensitivity in airway smooth muscle leading to enhanced inflammatory gene expression and cytokine secretion [99,100]. For type 17 inflammation, ILC3 populations are expanded and respond to IL-1β and IL-23 to secrete IL-17A and IL-22 [101]. Alveolar macrophages have been shown to secrete IL-1β and IL-23, suggesting contributions to expansion of ILC3 populations [102,103]. While less is known about how lung macrophages regulate ILC1 and ILC3 populations, their ability to produce cytokines associated with ILC1 and ILC3 differentiation highlight a potential role in orchestrating ILC activation in type 1 and 17 inflammation to influence asthma endotype and severity.

### 3.4. T Lymphocytes

Macrophages facilitate adaptive immunity through antigen presentation and cytokine secretion. Th cell responses are initiated by presentation of antigenic peptide by the major histocompatibility complex II (MHCII) on macrophages and interactions with the T cell receptor (TCR) [104]. Alveolar macrophages express MHCII and have been shown to be important antigen presenting cells (APCs) in the airway [105]. Macrophages can also influence the inflammatory milieu by acting as important drivers of Th1 and Th17 differentiation by secreting IL-12 and IL-6 [18,74,106,107]. Additionally, macrophages secrete type 1 and 17 cytokines IL-1β, TNF-α and IL-17A [108,109,110]. In cockroach allergen challenged mice, depletion of alveolar macrophages, via administration of dichloro-methylene-di-phosphonic acid disodium salt, reduced TNF-α levels in lung homogenates [110]. Much like the innate immune system, macrophages modulate Th cell responses that are most commonly associated with asthma and influence asthma endotype based on the inflammatory milieu.

Tissue-resident memory T cells (T_RM_) are a recently discovered T cell type that persist in the lung and, upon activation, secrete large amounts of inflammatory cytokines [111]. T_RM_ cells are found in barrier tissues in the lung, digestive, and reproductive tract to provide protection from pathogens [112,113]. Much like conventional CD4+ Th cells, these T_RM_ cells secrete Th1 cytokines IFN-γ and TNF-α, Th2 cytokines IL-4, IL-5, and IL-13, and Th17-associated cytokine IL-17A in large quantities [114,115,116]. In a mouse model of allergic airway disease via HDM challenge, Turner et al. found that CD4+ T_RM_ persisted in the lung 4–8 weeks post-allergen challenge [117]. Further, a short secondary HDM challenge that did not elicit a response in previously unchallenged mice (naïve) resulted in increased AHR and immune cell infiltration in mice challenged previously with HDM [117]. CD4+ T_RM_ were concluded to be the initiators of this heightened response, as increased AHR was observed during a secondary HDM challenge when circulating Th cells were depleted via anti-Thy1 antibody [117]. These studies provide evidence that T_RM_ cells are important upon re-exposure to an allergen and can be targeted to reduce the persistent inflammation associated with asthma. One striking difference between conventional Th cells and T_RM_ cells is that T_RM_ lack honing mechanisms to lymph nodes, so their activation is rapid locally within the lung [118,119]. This creates opportunities for macrophages to act as activating cells due to their proximal location to the airway by means of antigen presentation, co-stimulation, and cytokine secretion [120,121]. IL-15 is required for CD4+ T_RM_ generation and macrophages have been identified as potent producers of IL-15 in the lung [122,123]. Additional models of disease have identified the important interactions between macrophages and T_RM_. Cytokines secreted from macrophages are linked to the maintenance of T_RM_, not only in the lung, but also in other areas of the body such as the reproductive tract [124]. In a mouse model of herpes simplex virus 2 infection, Iijima et al. found that CCL5 was needed to support CD4+ T_RM,_ and macrophages served as the primary CCL5 source [124].

### 3.5. B Lymphocytes

B cells also have important roles in asthma pathogenesis and as a biomarker [125]. As part of humoral immunity, B cells secrete immunoglobulins (Igs), IgE and IgA [126,127]. IgE heightens type 2 responses by causing mast cell degranulation leading to increased IL-4 and IL-13 levels and bronchoconstriction [128,129]. IL-4 secreted from macrophages help drive IgE production by B cells, thus enhancing type 2 immune responses [130]. IgE has also been shown to have effects on macrophage function. Macrophages express FcεR and readily respond to IgE-mediated allergic airway inflammation [131]. In a mouse model of allergic airway inflammation via ovalbumin-specific IgE sensitization/challenge, depletion of alveolar macrophages using 2-chloroadenosine reduced immune cell infiltration, type 2 cytokine levels, mucous cell abundance, fibrosis, and airway smooth muscle mass [131]. In a study by Pellizzari et al., macrophages were isolated from human blood and differentiated into pro-inflammatory or anti-inflammatory phenotypes and incubated with IgE. Treatment with IgE increased macrophage TNF-α and IFN-γ production [132]. Although IgE has previously been associated with enhanced type 2 responses, the study by Pellizzari et al. suggests that IgE can induce macrophages to produce type 1 cytokines associated with severe asthma and corticosteroid insensitivity, indicating an importance of macrophages in determining asthma endotypes [9,100].

IgA is the primary Ig in the respiratory tract [133]. Decreases in IgA levels are associated with increases in airway bacterial colonization [87,134]. In a study by Ladjemi et al., bronchial epithelial cells were isolated from asthma patients and cultured with IL-4 and IL-13 [135]. IL-4 and IL-13 were found to downregulate polymeric immunoglobulin receptor (pIgR) at the site of IgA binding on airway epithelial cells, suggesting that IgA levels may be reduced in asthma [135]. Reduction in protective IgA levels creates the possibility for opportunistic pathogens to infiltrate the airways of asthmatics, resulting in infection, as discussed in a later section. As in the case of IgE, type 2 cytokines secreted by macrophages could reduce pIgR, leading to reduced IgA levels in the airways. In fact, depletion of macrophages using myeloid-specific Diphtheria Toxin A-driven apoptosis in a mouse model of muco-obstructive lung disease resulted in higher BAL levels of IgA, pointing to the possible role of dysfunctional macrophages negatively impacting host defense mechanisms [136].

In inflammatory conditions, B cells along with T cells form lymphoid structures known as inducible bronchus-associated lymphoid tissue (iBALT) that is embedded in lung tissue and enacts local immune responses, similarly to lymph nodes [137,138,139]. We have previously shown the increased formation of iBALTs in a mouse model of severe allergic airway inflammation accompanied by neutrophilic inflammation. Further, these iBALTs persisted in the presence of corticosteroids [140]. In a mouse model of fine particle exposure, Kuroda et al. found that IL-1α release from apoptotic alveolar macrophages promoted iBALT formation in the lungs of mice exposed to either aluminum oxide or silica particles [141]. IL-1α is increased in the BAL of asthma patients with neutrophilia compared to a more type 2-associated asthma [142]. In a mouse model of muco-obstructive lung diseases similar to cystic fibrosis (*Scnn1b*-Tg+ mice), depletion of macrophages via myeloid-specific Diphtheria Toxin-A induced apoptosis resulted in increased number of iBALTs and increased BAL Ig levels [136]. This was attributed to the need for enhanced adaptive immune system compensation to combat bacterial infection in the absence of macrophages. iBALTs have also been shown to cause macrophages to secrete matrix metalloproteinase 12 (MMP12) in a mouse model of COPD, which could possibly enhance airway remodeling by macrophages, as discussed in the next section [143]. Further investigation into interactions between macrophages and B cells will provide insight not only into important immune responses but the possible connection to persistent airway remodeling that is observed in patients with more severe asthma.

## 4. Pathological and Functional Changes in Asthma

Persistent airway remodeling refers to structural changes in the lung and is a key pathological feature in airway disease [22]. In asthma, the airway epithelium and smooth muscle undergo structural alterations, such as hyperplasia and hypertrophy, that contribute to airflow obstruction [144,145]. This section will discuss the current literature on how interactions between lung macrophages and airway structural cells promote pathological and functional changes in asthma.

### 4.1. Airway Epithelium

The airway epithelium lines the conducting airways and provides a protective barrier uniquely designed to combat invading pathogens that infiltrate the airways [146]. This innate barrier is composed of club cells, mucus-producing goblet cells, and ciliated cells that work in concert to remove unwanted pathogens, such as allergens, viruses, and bacteria [147,148]. The integrity of the airway epithelium is held together by tight junctions that attach to neighboring cells [149]. Exposure to allergen and chronic inflammation in asthma has been shown to damage epithelial tight junctions, increasing the accessibility of allergens, viruses, and bacteria to penetrate further into the lung [150,151,152]. Macrophages may have a direct connection with tight junctions during both homeostasis and inflammatory conditions [153]. In a mouse model of lung injury via lipopolysaccharide (LPS) administration, alveolar macrophages and epithelial cells worked in concert to reduce cytokine release, attenuate inflammation, and maintain epithelial barrier integrity [153]. Not much else is known about crosstalk between alveolar macrophages and airway epithelium in the context of asthma [19,154]. Future studies should focus on the protective functions of macrophages to maintain airway epithelial structural integrity in order to decrease accessibility of allergens, viruses, and bacteria.

Mucus is a vital contributor to the overall health of the lung by trapping inhaled pathogens and particles to keep them away from the lower airways [155]. Mucus is produced by goblet cells located within the airway epithelium and submucosal glands in larger airways [156]. Excessive accumulation of mucus in the airway is a prominent feature in asthma and in severe cases has been linked to the major cause of fatal asthma [156,157,158]. Increased mucus can be attributed to increases in the number of goblet cells and amount of mucus they produce [156]. Bronchial biopsies revealed a higher composition of goblet cells compared to healthy controls [159]. Further, patients with fatal asthma exhibited a significantly higher amount of goblet cells in the airway epithelium compared to asthma patients dying due to other causes, highlighting the severity of mucus accumulation in asthma [160]. IL-13 is a key inducer of mucus production and the increased abundance of goblet cells in asthma [156]. Macrophages are identified as a key source of IL-13 in airway disease [161]. Interstitial macrophage-derived IL-10 has been shown to reduce epithelial mucous cells’ abundance. HDM-challenged *Il10^−/−^* mice exhibited increased epithelial mucous cell abundance compared to HDM-challenged wild type (WT) mice. Further, adoptive transfer of WT interstitial macrophages into HDM-challenged *Il10^−/−^* mice reduced epithelial mucous cell abundance, suggesting a protective role of interstitial macrophages in regard to mucous cell abundance [43]. The ability of macrophages to both contribute to and reduce mucous cell abundance may lie in the subset of macrophage and in-depth investigation is needed to distinguish their roles in regulating mucous cell abundance and mucus production.

### 4.2. Airway Smooth Muscle

Airway smooth muscle (ASM) is a key contributor to thicker and stiffer airways, increased contraction, and airway narrowing. Airway hyperresponsiveness (AHR) is a feature in asthma and defined by increased airway smooth muscle force during contraction and airway narrowing in response to bronchoconstrictor histamine or acetylcholine [162]. Inflammatory cytokines, such as TNF-α and IL-1β, increase ASM proliferation and contractile force to restrict airflow in asthma [163,164]. Classically-activated macrophages are known producers of both TNF-α and IL-1β in asthma [103,165]. Aside from cytokine secretion, interactions between macrophages and ASM in the context of asthma are still poorly understood. Yet some insight can be drawn from co-culture experiments in atherosclerosis models. In a study by Butoi et al., human primary aortic smooth muscle cells were co-cultured with monocyte-like line THP-1 cells that were stimulated towards classical activation. Co-culture with THP-1 cells increased growth factors’ production by aortic smooth muscle cells. Conversely, responses from aortic smooth muscle cells increased matrix metalloproteinase (MMP)-9 and IL-1β in THP-1 cells, indicating a cross-talk between the two cell types [166]. Although this study was performed in aortic smooth muscle cells, one can speculate that similar mechanisms between airway smooth muscle and lung macrophages may be important in asthma.

### 4.3. Airway Fibrosis and Extracellular Matrix Deposition

Fibroblasts are structural cells that contribute to extracellular matrix (ECM) organization [167]. Matrix metalloproteinases (MMP) degrade ECM protein (collagens, fibronectin), regulating turnover and composition in the lung. Increased MMP activation contributes to inflammation and fibrosis through immune cell infiltration and ECM protein turnover [168,169]. In a study by Mautino et al., alveolar macrophages from people with asthma released higher MMP-9 levels compared to healthy controls [170]. Further, higher MMP-9 secretion from alveolar macrophages isolated from asthmatics was associated with increased airway thickness and faster decline in forced expiratory volume in one second (FEV_1_) [169].

Pro-fibrotic mediators activate fibroblasts to secrete extracellular matrix proteins, such as collagen and fibronectin, that stiffen airways [171]. Alveolar macrophages produce pro-fibrotic mediators that activate fibroblasts to promote airway remodeling [172,173]. PDGFs, PDGF-AA and PDGF-BB, are growth factors that induce collagen synthesis in fibroblasts [23]. In a study by Lewis et al., primary fibroblasts isolated from bronchial biopsies of patients with severe or mild-moderate asthma were cultured with PDGF-AA or PDGF-BB [23]. Fibroblasts isolated from severe asthmatics exhibited increased procollagen I expression compared to both healthy control and mild-moderate asthma groups and this was correlated with FEV_1_ decline [23]. In addition to alveolar macrophages, recruited macrophages have also been shown to enhance airway remodeling [77,174,175]. Asthmatics with increased bronchial wall thickening had higher levels of CCL2 compared to asthmatics with no thickening of the bronchial wall, indicating a role for the CCL2-CCR2 monocyte recruitment axis [175].

### 4.4. Airway Nerve Innervation

Nerves have integral roles in airway tone regulation, inflammation, and asthma [176]. Nerves secrete inflammatory mediators and bronchoconstrictors that induce smooth muscle contraction and contribute to eosinophil recruitment and activation [177,178]. Their importance is highlighted by the use of long acting anti-muscarinic drugs to induce bronchodilation in asthma [179]. Macrophages may contribute to increased airway nervous innervation leading to enhanced airway contraction and restricted air flow. One possibility lies in the macrophage’s role in eosinophil recruitment through IL-5 secretion. Studies show that IL-5-overexpressing mice exhibit increased eosinophilic infiltration, airway hyperresponsiveness, and airway nervous innervation [177,180]. Macrophages may also actively contribute to nerve growth and survival through neurotrophins (NT) secretion including NT3, nerve growth factor (NGF), and brain-derived neurotrophic factor (BDNF) during allergen challenge [181,182].

## 5. Inflammatory Resolution

As discussed in previous sections, the inflammation associated with asthma results in enhanced immune responses and structural changes that, if not mitigated, lead to decreased lung function and poorer quality of life for people with asthma [183]. Inflammatory resolution is essential for lung restoration and homeostasis after an asthma exacerbation [183]. Macrophages have been found to play a critical role in resolution of lung inflammation by clearing apoptotic cells, via efferocytosis, and promoting tissue repair (Figure 2) [6,64,184]. This section will discuss different mechanisms macrophages use to promote tissue repair and resolve inflammation.

### 5.1. Efferocytosis

Apoptotic eosinophils and neutrophils are common in asthma [185]. Efferocytosis is a process that removes apoptotic cells during the inflammatory resolution stage and is essential for returning the lung to homeostasis [186]. Defective efferocytosis has been linked to secondary necrosis and release of damage signals that further enhance inflammation [185,187]. Macrophages are premier cell types for efferocytosis, but this mechanism is found to be dysfunctional in asthma, particularly in macrophages from severe asthmatics [188,189]. In a study by Simpson et al., macrophages isolated from the sputum of patients with severe asthma exhibited decreased efferocytosis of apoptotic bronchial epithelial cells compared to macrophages isolated from mild-moderate asthmatics [189]. Increases in oxidative stress in macrophages are associated with classical activation of macrophages and reduced ability to remove apoptotic cells [20]. The presence of these classically-activated macrophages is increased in people with severe asthma as compared to mild-moderate asthma [190], which may explain the decrease in efferocytosis. In a study by Ryan et al., impaired efferocytosis was linked to increases in oxidative stress in COPD lung macrophages [191]. Restoring NRF2, a key regulator of antioxidant responses and oxidative stress, expression in macrophages reduced oxidative stress and restored efferocytosis [191]. Although this study was performed in a model of COPD, NRF2 expression has also been found to be reduced in asthma macrophages and could contribute to impaired efferocytosis and asthma severity [192]. These data would suggest that antioxidant mechanisms may be important for reducing oxidative stress and enhancing macrophages’ efferocytosis activity in asthma.

In addition to clearing apoptotic cells, efferocytosis-related mechanisms may also promote macrophage-mediated tissue repair. A recent study has shown that efferocytosis of apoptotic cells also induces macrophages to secrete IL-10 and TGF-β to help resolve and repair damaged tissue [193]. In summary, macrophages play a key role in clearing apoptotic cells and contributing to inflammatory resolution. Targeting dysfunctional macrophage efferocytosis in asthma, particularly more severe endotypes, may provide a means to reduce persistent inflammation caused by uncleared apoptotic cells.

### 5.2. Tissue Repair

During asthma, environmental insults and infection causes injury to airway structural cells, inducing inflammation and compromising their structural integrity. Tissue repair mechanisms help resolve inflammation and restore the airway integrity. However, impaired repair and resolution mechanisms can lead to airway fibrosis which can cause airway narrowing and obstruction of airflow [194,195,196]. Several studies have shown that macrophages contribute to the repair of the airway epithelium, airway smooth muscle, and fibrosis.

Damage to the airway epithelium during allergic airway inflammation results in increased mucous cells and expression of damage signals that enhance inflammation and increase lung tissue permeability [196]. Epidermal growth factor (EGF) is key in epithelial cell proliferation and repair and has been shown to be produced by macrophages [197,198]. The epithelial repair process has mainly been tied to alternatively activated macrophages [199]. Using a mouse model of lung injury via naphthalene administration that resulted in a loss of club cells within the epithelium, Dagher et al. demonstrated that epithelial regeneration post-naphthalene insult required resident lung macrophages that were skewed towards the alternatively-activated phenotype. Further, this repair process was linked to IL-33 macrophage stimulation, a driver of alterative activation [200], as naphthalene-challenged *St2^−/−^* mice exhibited incomplete epithelial repair. Adoptive transfer of WT macrophages into naphthalene-challenged *St2^−/−^* mice restored epithelial repair following injury [201].

Macrophages, particularly alternatively-activated macrophages, have been shown to reduce airway fibrosis in animal models. In a study by Li et al., mice were sensitized/challenged with ovalbumin intranasally for 7 days, then treated with exosomes isolated from alternatively-activated bone marrow derived macrophages on day 6, 7, and 8 [202]. Ovalbumin-challenged mice not treated with exosomes exhibited increased fibrosis and peribronchiolar immune cell aggregation. Conversely, treatment with alternatively-activated macrophage exosomes significantly decreased both fibrosis and immune cell aggregation. This mechanism was found to involve exosomes carrying miR-370, which has previously been associated with mitigation of LPS-induced lung injury [202,203]. This study highlights the role of macrophages in repairing fibrosis during the resolution of inflammation.

## 6. Host Defense against Pathogens

The confounding effects of airway remodeling, such as epithelial permeability and mucus stasis, enhance susceptibility to viruses and bacteria [204]. These infections result in an altered immune environment that may exacerbate asthma symptoms and affect asthma severity [205,206]. This section will discuss the role of macrophages during viral and bacterial infections associated with asthma (Figure 3).

### 6.1. Viral Infection

Viral infections, most notably rhinovirus, are the primary cause of acute exacerbations in both children and adults with asthma [205,207]. Rhinovirus is demonstrated to induce robust Th2-associated responses that amplify asthmatic symptoms [208,209]. There is increasing evidence that viral infection enhances macrophage inflammatory responses in asthma [210,211,212]. In an ovalbumin mouse model of allergic asthma, rhinovirus infection further increased type 2 cytokine (e.g., CCL11, IL-4, and IL-13) production, eosinophil and neutrophil infiltration, and AHR [210]. In this study, rhinovirus was found to localize with lung macrophages and clodronate macrophage depletion suggested that lung macrophages play an integral role in the heightened eosinophil infiltration and AHR in ovalbumin-challenged and rhinovirus-infected mice [210]. Interestingly, RNA-seq analyses revealed that rhinovirus infected human monocyte-derived macrophages polarized to M1-like conditions with IFN-γ in vitro exhibited a robust inflammatory response related to type 1 inflammation [212]. Conversely, infected human monocyte-derived macrophages, polarized to M2-like conditions with IL-4 in vitro, had a very limited response [212]. These findings suggest rhinovirus has differential effects on macrophages polarization, but additional studies are needed to determine how macrophages enhance the type 2 response upon rhinovirus infection in allergic asthma.

Similar to rhinovirus, respiratory syncytial virus (RSV) is associated with type 2 inflammatory responses and acute exacerbation, particularly in infants and young children [213]. Inflammatory responses to RSV are induced in lung macrophages and are important for host immunity against RSV [214,215]. Upon RSV infection, alveolar macrophages up-regulate pro-inflammatory mediator production that includes type I interferon and antiviral responses through mitochondrial antiviral (MAV) and retinoic acid-inducible gene (RIG) signaling [21,216,217]. These important antiviral pathways contribute to inflammatory monocyte lung recruitment for RSV clearance in mice [216,218]. The number of circulating monocytes is increased in RSV infected patients and is needed for effective defense against RSV. Yet reduced human monocyte leukocyte antigen (HLA)-DR expression and IL-10 induction was found to be associated with severe RSV infection [219]. Similarly, alveolar macrophages are important for host defense against RSV, as macrophage depletion in mice show increased viral load in RSV infected mice [220,221]. While RSV is strongly linked to type 2 responses and asthma risk, less is known about how RSV affects type 2-associated responses in lung macrophages. Recent studies in mice showed RSV infection increases IL-33 mRNA expression in alveolar but not interstitial lung macrophages [222]. Another study has reported enhanced alternatively activated macrophage populations that produce IL-4 and IL-13 during RSV infection in mice [223]. These data indicate that recruited monocytes and alveolar macrophages are important for host defense against RSV but may also produce type 2 inflammatory cytokines that contribute to asthma pathogenesis.

Rhinovirus has also been implicated in reducing macrophage phagocytic activity, with rhinovirus infected COPD alveolar and monocyte-derived macrophages having impaired ability to phagocytose *Streptococcus pneumoniae* and *Haemophilus influenzae* (NTHi) [224]. Similar effects on phagocytosis were observed in alveolar and monocyte-derived macrophages treated with the TLR3 agonist, Poly I:C, which also dampened IL-10 production [224]. Macrophage phagocytosis in the context of RSV infection is also important. In addition to pro-inflammatory mediator production, RSV infection increases alveolar macrophages phagocytosis, via the Fc-γ receptor [225,226,227]. However, studies using RSV co-infection with bacteria suggest that RSV infection can reduce macrophage bacterial phagocytic activity [228]. While the effect of rhinovirus and RSV on macrophage host defense responses in asthma have yet to be explored, these findings suggest that rhinovirus and RSV infection can alter pro-inflammatory responses and phagocytosis in lung macrophages, which may contribute to airway inflammation in asthma.

### 6.2. Bacterial Dysbiosis and Colonization

The lung microbiome consists of numerous bacterial species that are important for lung health and immunity [229]. Airway bacteria composition is altered in asthma with implications for airway inflammation and asthma severity [17,230,231]. A recent study using 16S rDNA V4 region sequencing found differences in α and β diversity between asthma subjects with low and high eosinophils, suggesting a link between microbial composition and type 2 inflammation [230]. A similar study that analyzed sputum samples in a severe asthma cohort found that increased *Actinomycetaceae* abundance correlated with greater eosinophil counts in asthma patients [232]. Differences in bacterial composition in the airway are also observed among severe asthma endotypes with neutrophil or mixed granulocyte infiltration [233]. These data suggest that the airway microbiome may influence asthma pathogenesis and impact asthma management.

In addition to changes to airway microbiota composition, severe asthma is associated with increase colonization of pathogenic bacteria species [234]. Increased *Staphylococcus aureus*, non-typeable *Haemophilus influenzae* (NTHi), and *Moraxella catarrhalis* colonization are often found in asthmatic airways and associated with worsened outcomes [206,235,236,237]. Here, macrophages may play a key role in the emergence of these species and affect their impact on asthma pathogenesis. Intranasal NTHi inoculation prior to ovalbumin sensitization and challenge in mice reduces type 2 responses but increases Th17-associated responses with increased neutrophil infiltration and IL-17A levels [16,238]. These inflammatory responses are also resistant to corticosteroids [238], which is consistent with the observed increases in NTHi colonization in individuals with severe asthma [239,240]. Further, recent studies show that NTHi induces a transcription profile in human monocyte-derived macrophages that reflects pro-inflammatory response promoting neutrophil recruitment and activation [235,241]. Liang et al. showed that bronchoalveolar lavage macrophages isolated from people with severe asthma exhibit decreased *Haemophilus influenzae* and *Staphylococcus aureus* phagocytosis compared to healthy controls [242]. These data suggest that defective phagocytic activity in alveolar macrophages lead to increased bacterial colonization that could subsequently lead to persistent airway inflammation, which is insensitive to corticosteroids.

While the role of bacterial colonization in asthma exacerbation is not fully established [243] (as opposed to COPD), treatment with azithromycin, an antibiotic macrolide, reduces exacerbation frequency and improves quality of life among people with asthma [244]. In a study by Gibson et al., people with poorly controlled asthma that were also on a medium-to-high corticosteroid dose were treated with either placebo or azithromycin three times a week for 48 weeks [244]. Compared to the placebo group, patients receiving azithromycin exhibited a significant reduction in exacerbations, cough, and sputum production [244]. Although there were beneficial outcomes with azithromycin treatment, there were noted adverse side effects with patients in the azithromycin group [244]. Taylor et al. revealed that people with poorly controlled asthma on a 48-week azithromycin treatment regimen experienced no significant reduction in sputum bacterial load [245]. Although antibiotic treatment reduced *Haemophilus influenzae*, microbes such as *Staphylococcus pneumoniae*, *S. aureus*, *Pseudomonas aeruginosa*, and *Moraxella catarrhalis* were not impacted [245].

Azithromycin has been shown to enhance macrophage phagocytic activity and affect macrophage polarization by decreasing gene expression associated with classical activation while increasing expression of alternative activation genes [246,247,248]. Alternative polarization induced by azithromycin was demonstrated to involve suppression of NFκB and Stat1 signaling [249]. This suggests that azithromycin can modulate pro-inflammatory responses that are involved in classical macrophage activation and neutrophilic inflammation in severe asthma.

Despite bactericidal and anti-inflammatory effects, long term use of antibiotics has been linked to the generation of antibiotic resistant bacterial strains in severe asthma [245,250,251], potentially limiting their utility. The possible generation of antibiotic resistant microbes has elicited a call to reduce overuse of antibiotics and pursue therapeutic avenues that involve enhancing host defense to invading microbes. Given the importance of macrophages recognizing these unwanted invaders and their powerful bactericidal activity, they may be an attractive target for reducing bacterial load in people with asthma. For example, novel nonantibiotic macrolides, which are an alternative to antibiotic macrolides, were recently shown to enhance phagocytic activity and induce anti-inflammatory responses in primary alveolar macrophages [252].

## 7. Macrophages Contribute to Severe Asthma Endotypes

Asthma is a heterogenous lung disease that possesses several endotypes that are though to influence sensitivity to corticosteroids, an anti-inflammatory drug commonly used in asthma [253]. Broadly, severe asthma can be classified based on sputum inflammatory cell compositions as eosinophilic, neutrophilic, mixed granulocytic (eosinophils and neutrophils), or pauci-granulocytic (absence of granulocytes) [254]. Macrophages have been found to contribute to the pathogenesis of these severe endotypes. In neutrophilic asthma, Fricker et al. demonstrate that macrophages isolated from people with neutrophilic asthma exhibit enriched pathways for neutrophilic migration, inflammatory responses, bacterial responses, and lymphocyte activation when compared to macrophages isolated from non-neutrophilic asthma. These data suggest that macrophages from neutrophilic asthma are transcriptionally altered to be more pro-inflammatory [255]. Macrophages are also found to be important in pauci-granulocytic asthma. Olgac et al. reported that people with pauci-granulocytic asthma possess higher macrophage numbers in the sputum when compared to people with eosinophilic, neutrophilic, or mixed granulocytic asthma [256]. This pauci-granulocytic endotype was also more poorly controlled compared to other groups in improving FEV_1_ with treatment [256].

In summary, macrophages are associated with the different endotypes of asthma as well as with their contributions to the efficacy of corticosteroids. These studies form the foundation for future investigations into identifying macrophage targeted therapies to reduce inflammation and increase corticosteroid efficacy. One strategy is reprogramming of macrophages to the alternatively activated state to promote inflammatory resolution [257,258]. However, this avenue needs to be approached with caution. As outlined in this review, alternatively activated macrophages, particularly uncontrolled alternative activation, can itself lead to enhanced eosinophil recruitment and release of pro-remodeling mediators that can further exacerbate asthma. This may be accomplished by antioxidant therapies. During asthma and associated oxidative stress, macrophages are classically activated and antioxidant therapies reducing reactive oxidant species could skew macrophages to the pro-resolution stage earlier in allergic airway inflammation.

## 8. Knowledge Gaps and Conclusions

Macrophages play a critical role, not only in asthma pathogenesis, but also in return to homeostasis post-allergic inflammation. Further, lung macrophages contribute to asthma phenotype, severity, and may affect corticosteroid sensitivity [82]. Identifying specific roles of each macrophage subset in asthma could allow for cell-specific therapeutic targets that may reduce airway immune cell infiltration, bacterial infection, and airway remodeling leading to improved outcomes. For instance, alveolar macrophages have been identified as key recruiters of inflammatory immune cells but are also vital for pathogen clearance. This subset is also extremely critical for the resolution of the inflammatory response post-allergen exposure [5]. Harnessing the antibacterial capabilities of alveolar macrophages during infection and pro-resolution mechanisms may prove beneficial in reducing inflammation associated with asthma. Secondly, interstitial macrophages have been found to reduce neutrophilic infiltration that is common in more severe cases of asthma [43]. Utilizing this capability may further reduce the increased asthma severity associated with neutrophilic infiltration and corticosteroid sensitivity. Third, recruited monocytes are identified as proinflammatory in asthma and reduction in these populations through interference with the CCL2-CCR2 chemotactic axis may alleviate allergic airway inflammation. Recent advances in single-cell RNA-sequencing (scRNA-seq) reveal distinguishable differences in transcriptional profiles and possible functions that further differentiate macrophage subsets will allow for more in-depth investigation into their differential roles in asthma [37,259]. Utilizing this technology will allow for identification of novel pathways associated with each subset and could help identify possible novel therapies to mitigate allergic airway inflammation.

There are knowledge gaps however when it comes to the role of macrophages in asthma. Although experiments investigating cross-talk between macrophages and immune cells and structural cells have been performed, much remains unknown. For example, T_RM_ are a newly characterized cell type that persist in the tissue [116]. Given the proximity of macrophages to these cytokine super-secretors, it is possible that macrophages play a huge role in T_RM_ activation upon re-exposure to allergen. Other than cytokine production, not much is known about direct interactions between macrophages and airway smooth muscle cells despite their close proximity. As seen with direct contact interactions with airway epithelium [260], there could possibly be physical interactions that result in increased ASM contraction and/or stiffness, leading to increased airway hyperresponsiveness.

Phagocytosis and efferocytosis are key mechanisms that have been found to be impaired in asthma macrophages [53]. This creates an environment for pathogens to thrive, as well as uncleared apoptotic cells releasing damage signals to further enhance inflammation [185]. Identifying key mechanisms to enhance macrophage phagocytosis and efferocytosis function will increase host defense and also reduce inflammation caused by uncleared apoptotic cells.

In summary, macrophages are an important cell type that contribute to both harmful and beneficial processes in asthma. Harnessing their power will be important to reduce exacerbation and improve overall quality of life in people with asthma.

## Figures and Tables

**Figure 1 ijms-24-10451-f001:**
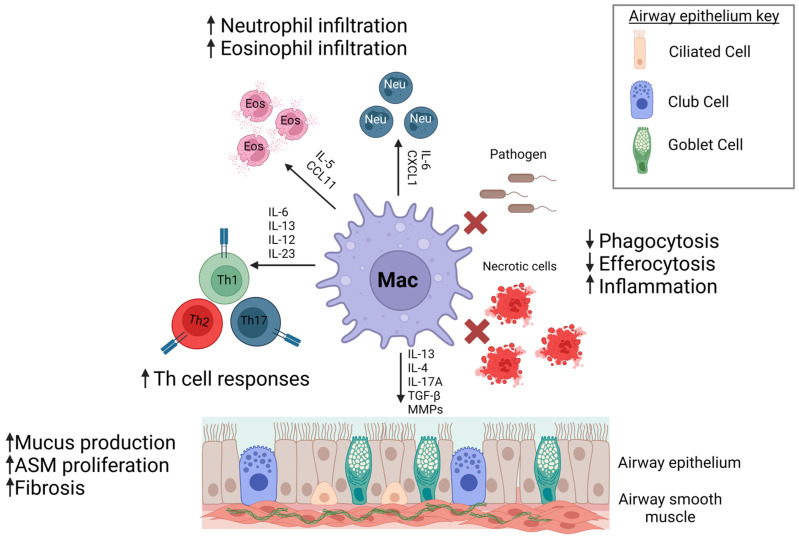
Macrophages contribute to asthma pathogenesis. Dysfunctional macrophages in asthma can contribute to enhanced immune cell responses, increased remodeling, and decreased clearance of inhaled pathogens and dead cells leading to persistent inflammation and asthma severity. Created with Biorender on 13 June 2023.

**Figure 2 ijms-24-10451-f002:**
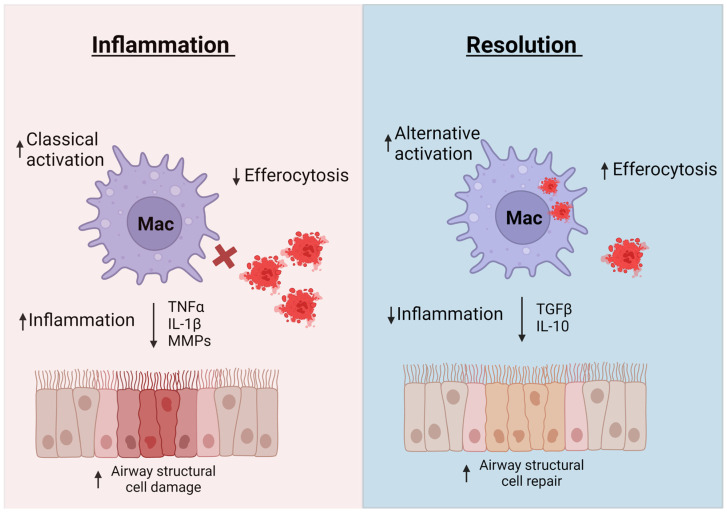
Macrophages contribute to both inflammation and inflammatory resolution. In inflammatory conditions, macrophages produce mediators such as TNF-α, IL-1β, and MMPs that lead to lung structural cell damage and remodeling. Dysfunctional efferocytosis leads to persistence of necrotic cells in lung tissue that release damage signals and further enhance inflammation. Macrophages contribute to inflammatory resolution by secreting mediators such as IL-10 and TGFβ, to repair lung structural integrity and by engulfing apoptotic cells to reduce inflammation. Created with Biorender on 13 June 2023.

**Figure 3 ijms-24-10451-f003:**
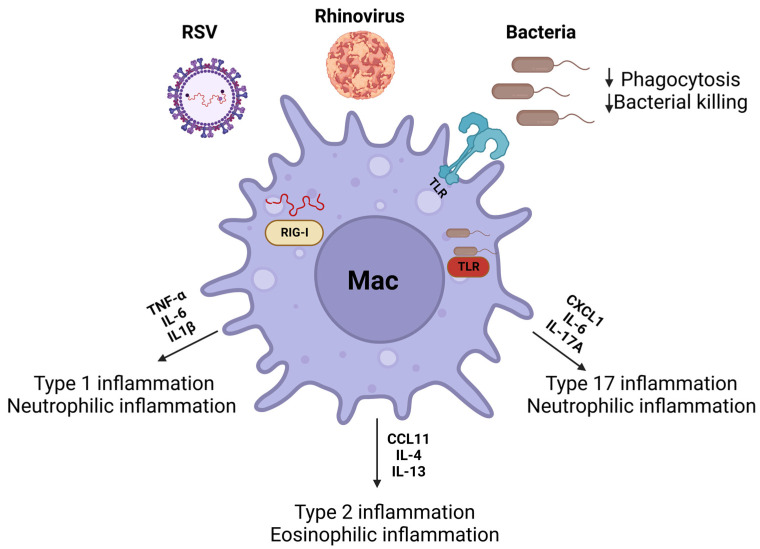
Macrophages play a key role in removing inhaled viruses and bacteria from the airways. Viruses (rhinovirus and RSV) and bacteria (*Staphylococcus aureus*, non-typeable *Haemophilus influenzae*, and *Moraxella catarrhalis*) can augment macrophage polarization to heighten type 1, 2, and 17 inflammation. These responses are mediated through intracellular (RIG-1, TLR3, and TLR9) and extracellular (TLR2 and TRL4) pattern recognition receptors. Created with Biorender on 13 June 2023.

## Data Availability

Not applicable.

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
