# Peer review of "Macrophages Orchestrate Airway Inflammation, Remodeling, and Resolution in Asthma"

_ijms, 2023, doi:10.3390/ijms241310451_

Round 1
Reviewer 1 Report
The review 'Macrophages orchestrate airway inflammation, remodeling, 2 and resolution in asthma' by Britt et al is a nicely written review and covers major aspects related to the topic. I have the following comments:
-Please include IL-17 in Fig 1 as it is also secreted by lung macrophages, e.g. PMID: 18941201; PMID: 37119678.
-Please add goblet cells in Fig. 1.
-Please add a section on the role of macrophages in corticosteroid insensitivity in severe or mixed granulocytic asthma.
-Line 251-265, please explain the role of macrophages in these studies, otherwise delete this paragraph as it talks about only T cells in asthma.
-'Macrophages expression FcεR and readily respond in IgE-mediated allergic airway inflammation'; 'and difficulty breathing'; please correct grammar.
Author Response
-Please include IL-17 in Fig 1 as it is also secreted by lung macrophages, e.g. PMID: 18941201; PMID: 37119678.
Response: IL-17A has been added to figure 1
-Please add goblet cells in Fig. 1.
Response: A key has been inserted to identify airway epithelial cells.
-Please add a section on the role of macrophages in corticosteroid insensitivity in severe or mixed granulocytic asthma.
Response: A section detailing the current literature on the role of macrophages to asthma endotypes and corticosteroid sensitivity has been added.
-Line 251-265, please explain the role of macrophages in these studies, otherwise delete this paragraph as it talks about only T cells in asthma.
Response: As a newly described T cell subset, we feel it is important to highlight these cells as major producers of cytokines and possible key players in asthma. In our opinion, the interactions between lung tissue resident memory cells (TRM) and macrophages have been understudied so we commented on how macrophages may induce TRM activation through antigen presentation and cytokine secretion.
-'Macrophages expression FcεR and readily respond in IgE-mediated allergic airway inflammation'; 'and difficulty breathing'; please correct grammar.
Response: The grammatical errors have been corrected.
Reviewer 2 Report
The article ‘Macrophages orchestrate airway inflammation, remodeling, and resolution in asthma’ by Britt; Jr et al. summarized the current understanding of the roles of lung macrophages in airway remodeling and inflammatory response. The review article is structured nicely, and the work is relevant to the readership of ‘IJMS.’ However, sentences are not crafted carefully in several places throughout the manuscript. In addition, some other figures are required. The authors need to address the following concerns in the current version of the manuscript. The issues are listed below.
The Abstract is written in very casual language. Therefore, it needs to be rewritten.
There are many verb tense inconsistencies throughout the manuscript. Further, the writing style, typographical and grammatical errors should be corrected in the revised version of the manuscript.
It would be beneficial for the readers if sections 5 and 6 are summarized in additional figures.
While the authors have attempted to summarize the relevant research in the field, the new hypotheses, open questions, or strategies to tackle them should be emphasized a bit more.
There are many verb tense inconsistencies throughout the manuscript. Further, the writing style, typographical and grammatical errors should be corrected in the revised version of the manuscript.
Author Response
The Abstract is written in very casual language. Therefore, it needs to be rewritten.
There are many verb tense inconsistencies throughout the manuscript. Further, the writing style, typographical and grammatical errors should be corrected in the revised version of the manuscript.
Response: The manuscript has been revised to address grammatical and language issues.
It would be beneficial for the readers if sections 5 and 6 are summarized in additional figures.
Response: Suggested figures have been added. (figures 2 and 3)
While the authors have attempted to summarize the relevant research in the field, the new hypotheses, open questions, or strategies to tackle them should be emphasized a bit more.
Response: We have added additional material throughout the manuscript that addresses new ideas on how to target macrophage responses in asthma including:
- Section 7
Comments on the Quality of English Language
There are many verb tense inconsistencies throughout the manuscript. Further, the writing style, typographical and grammatical errors should be corrected in the revised version of the manuscript.
Response: The manuscript has been revised to address grammatical and language issues.
Reviewer 3 Report
The authors provide an in-depth summary of the different subtypes and roles of macrophages in the development of asthma.
Specific comments:
Introduction:
· - Lines 49-51, a broad introduction of asthma, should come before lines 44-48, which give a bit more detail of asthma phenotypes.
· - There is a large body of work on severe, steroid-resistant asthma, more of that should be mentioned here.
Section 2:
· - Lung macrophage subsets; again there is a huge body of literature on markers used to identify/define different human macrophages which is not adequately covered in this section. E.g markers such as MARCO, CD206, HLA-DR
General:
· - There is a lot of information in this review, and it would be beneficial for the reader to have another 1 or 2 schematics to break up the text a bit.
Author Response
Introduction:
- - Lines 49-51, a broad introduction of asthma, should come before lines 44-48, which give a bit more detail of asthma phenotypes.
- - There is a large body of work on severe, steroid-resistant asthma, more of that should be mentioned here.
Response: This has been added in introduction as well as throughout the manuscript (section 7).
Section 2:
- - Lung macrophage subsets; again there is a huge body of literature on markers used to identify/define different human macrophages which is not adequately covered in this section. E.g markers such as MARCO, CD206, HLA-DR
Response: A section on macrophage activation markers has been added.
General:
- - There is a lot of information in this review, and it would be beneficial for the reader to have another 1 or 2 schematics to break up the text a bit
Response: Suggested figures have been added in the form of Figures 2 and 3.
Round 2
Reviewer 2 Report
After careful examination of the revised manuscript, the response of the authors to previous reviews, and the changes made in the manuscript, I gather that the revised version of the manuscript has addressed the major concerns raised in the previous version of the paper. Overall, the article is now easy to read, and there is a logical interpretation of the data (with reasonable assumptions). However, there is still room for improvement in the writing style and fix verb-tense consistency errors in the manuscript. Hence, I endorse the publication of this paper pending minor language corrections.
There is still room for improvement in the writing style and fix verb-tense consistency errors in the manuscript. Hence, I endorse the publication of this paper pending minor language corrections.